# Induced Necroptosis and Its Role in Cancer Immunotherapy

**DOI:** 10.3390/ijms251910760

**Published:** 2024-10-06

**Authors:** Ziyao Zhang, Fangming Zhang, Wenjing Xie, Yubo Niu, Haonan Wang, Guofeng Li, Lingyun Zhao, Xing Wang, Wensheng Xie

**Affiliations:** 1The Key Laboratory of Organic-Inorganic Composites, Beijing Laboratory of Biomedical Materials, College of Life Science and Technology, Beijing University of Chemical Technology, Beijing 100029, China; 2023201328@buct.edu.cn (Z.Z.); 2023210972@buct.edu.cn (F.Z.); 2021110034@buct.edu.cn (Y.N.); 2020080009@buct.edu.cn (H.W.); ligf@mail.buct.edu.cn (G.L.); 2School of Chemistry and Chemical Engineering, Guizhou University, Guiyang 550025, China; xwj15761987183@163.com; 3State Key Laboratory of New Ceramics and Fine Processing, School of Materials Science and Engineering, Tsinghua University, Beijing 100084, China; lyzhao@mail.tsinghua.edu.cn

**Keywords:** necroptosis, immunogenic cell death, DAMPs, cancer immunotherapy, immune microenvironment

## Abstract

Necroptosis is a type of regulated cell death (RCD) that is triggered by changes in the extracellular or intracellular milieu that are picked up by certain death receptors. Thanks to its potent capacity to induce immunological responses and overcome apoptotic resistance, it has garnered significant attention as a potential cancer treatment. Basic information for the creation of nano-biomedical treatments is provided by studies on the mechanisms underlying tumor necroptosis. Receptor-interacting protein kinase 1 (RIPK1)–RIPK3-mediated necroptosis, Toll-like receptor domain-containing adapter-inducing interferon (IFN)-β (TRIF)–RIPK3-mediated necroptosis, Z-DNA-binding protein 1 (ZBP1)–RIPK3-mediated necroptosis, and IFNR-mediated necroptosis are the four signaling pathways that collectively account for triggered necroptosis in this review. Necroptosis has garnered significant interest as a possible cancer treatment strategy because, in contrast to apoptosis, it elicits immunological responses that are relevant to therapy. Thus, a thorough discussion is held on the connections between tumor cell necroptosis and the immune environment, cancer immunosurveillance, and cells such as dendritic cells (DCs), cytotoxic T cells, natural killer (NK) cells, natural killer T (NKT) cells, and their respective cytokines. Lastly, a summary of the most recent nanomedicines that cause necroptosis in order to cause immunogenic cell death is provided in order to emphasize their promise for cancer immunotherapy.

## 1. Introduction

Necroptosis (also called “programmed necrosis”) is a form of regulated cell death (RCD) initiated by perturbations of the extracellular or intracellular microenvironment detected by specific death receptors, including (but not limited to) receptor-interacting protein kinase 1 (RIPK1), tumor necrosis factor receptor 1 (TNFR1), Toll-like receptor domain-containing adapter-inducing IFN-β (TRIF) (including Toll-like receptors 3 (TLR3), TLR4), Z-DNA-binding protein 1 (ZBP1), and ICP6 [1,2,3]. Generally, the activation of death receptors initiates the formation of a cytosolic signaling platform for caspase-8 (Casp8, complex IIB) [4]. This complex encompasses RIPK1, Casp8, Fas-associated death domain (FADD), and cellular FLICE/Casp8 inhibitory protein (cFLIP). Notably, this complex exerts regulatory control over Casp8-dependent apoptosis and RIP3-dependent necroptosis. Consequently, Casp8 acts as a deterrent to programmed necrosis, potentially achieved through mechanisms such as the direct cleavage of RIPK1 and/or RIPK3, leading to the separation of their kinase and the RIP homotypic interaction motif (RHIM) domains, or by targeting another component within the pathway [5].

As malignant cells are often resistant to apoptotic RCD under conventional treatments [6], necroptosis has attracted great attention, for it is regarded as an alternative defense pathway in the infected cells when apoptosis is inhibited [7]. Necroptosis may have evolved as an antiviral defense mechanism because certain viruses, including cytomegalovirus, herpes simplex virus, vaccinia virus, and adenovirus, encode inhibitors of Casp8 to prevent apoptosis [8]. Particularly, inducing and/or manipulating necroptosis in anti-cancer therapies represents a promising therapeutic approach for bypassing acquired or intrinsic apoptosis resistance, serving as an alternative way to eliminate apoptosis-resistant cancer cells. Moreover, necroptosis holds particular significance for the immune system via releasing damage-associated molecular patterns by aiding in the removal of cells infected by viruses or bacteria. It also presents a close relationship with cancer immunosurveillance. Necroptosis can act as a “whistle blower” for the activation of the immune response and is regarded as a potential strategy for effective cancer immunotherapy.

Here, we critically make a summary of currently available advances in necroptosis, the potential pathways to induce cancer necroptosis, and how these processes regulate the immune response, proposing its significant role for cancer immunotherapy.

## 2. Milestone on Discovery of Necroptosis

A significant challenge in investigating the role of necroptosis in vivo has been the absence of a definitive molecular marker for the in situ identification of necroptotic cells (Figure 1). Luckily, continuous milestones have been achieved to disclose the molecular mechanisms of necroptosis during the past twenty years. In 2000, the protein kinase RIPK1 was identified as an essential regulator for necrotic cell death induced by tumor necrosis factor α (TNF-α), TNF-related apoptosis-inducing ligand (TRAIL), or Fas ligand (FasL) in a kinase-activity-dependent manner [9]. It thus became clear that death receptor-initiated necrosis is tightly regulated by the kinase activity of RIPK1. This form of death receptor-initiated RIPK1-dependent necrosis is referred to as “programmed necrosis” [10]. In 2005, the term necroptosis was introduced to describe this regulated necrosis when necrostatin-1 (Nec-1) was identified as a specific chemical inhibitor of TNF-induced necrosis [11]. Nec-1 was further found to inhibit the kinase activity of RIPK1, and this inhibitor has been used extensively for the detection of necroptosis and for the evaluation of the in vivo basic biological significance of necroptosis [12]. In 2009, the kinase RIPK3 was found to regulate necroptosis [13,14]. RIPK3 can associate with RIPK1 to form a protein complex (termed a necrosome) through the RHIM domains of both proteins. The formation of the necrosome activates RIPK3, and this process leads to the phosphorylation of RIPK3. The discovery of RIPK3 functioning as key regulator of TNF-induced necrosis is the key step to unraveling the necroptosis pathway [13]. In 2011, TRIF was found to form a complex with RIPK3 under TLR3/TLR4 activation, which was essential for TLR3/TLR4-mediated necroptosis, although the relative contribution of RIPK1 to this process has not been fully dissected [15]. In 2012, a mixed-lineage kinase domain-like pseudokinase (MLKL) was found to be a downstream substrate of RIPK3, and its phosphorylation by RIPK3 was confirmed to be essential for necroptosis execution [16]. The RIPK3-mediated phosphorylation of MLKL activates MLKL, leading to MLKL oligomerization and membrane translocation for the execution of necroptosis [17]. In 2015, the herpes simplex virus type 1 (HSV-1) protein ICP6 was found to trigger necroptosis via interacting with RIPK1/RIPK3 through its RHIM domain and forming dimers/oliogmers [18]. In 2016, the discovery revealed that RIPK1 acts as an inhibitor of skin inflammation by impeding the ZBP1-mediated activation of RIPK3/MLKL-dependent necroptosis. In cells expressing RIPK1, the RIPK1 receptor-interacting RHIM was observed to strongly interact with phosphorylated RIPK3, indicating that the RIPK1 RHIM has the capability to hinder the binding of ZBP1 and the subsequent activation of RIPK3 [3]. In 2022, a study demonstrated that the depletion or mutation of adenosine deaminase RNA specific 1 (ADAR1), a major repressor of immune responses activated by endogenous retroviral elements, would lead to Z-RNA accumulation and ZBP1 activation, which culminated in RIPK3-mediated necroptosis [19]. ADAR1 suppresses intrinsic Z-RNAs and reveals ZBP1-mediated necroptosis as a novel factor influencing tumor immunogenicity that is obscured by ADAR1. In 2023, researchers reported that the defective process of prelamin A will trigger nuclear RIPK1-dependent necroptosis and inflammation. This study suggests RIPK1 as a feasible target for prelamin A-associated progeroid disorders [20].

Necroptosis is a biological process meticulously regulated by a variety of proteins and regulatory factors. For instance, cellular Inhibitors of Apoptosis (cIAPs) suppress the initiation of necroptosis through their interaction with RIPK1, and the X-linked Inhibitor of Apoptosis Protein (XIAP), serving as an E3 ubiquitin ligase, negatively regulates RIPK1 and RIPK3, thereby inhibiting necroptosis [21]. Under specific conditions, the activation of Casp8 can lead to the cleavage of RIPK1, blocking the progression of necroptosis [4,22]. The Extracellular Signal-Regulated Kinase signaling pathway negatively regulates necroptosis by promoting the expression of cIAPs [23]. Glucocorticoids inhibit necroptosis by activating specific signaling pathways. Cyclin-dependent kinases regulate necroptosis by affecting the phosphorylation status of RIPK1 [24]. The tumor suppressor protein p53 can influence necroptosis through various mechanisms, including the regulation of RIPK1 and RIPK3 expression, and Bcl-2 family proteins, such as Bcl-2 and Bax, modulate necroptosis through their interactions with RIPK1 [25,26]. Additionally, metabolic products in the tumor microenvironment, such as ATP, Reactive Oxygen Species (ROS), and fatty acid derivatives, can also regulate cellular necroptosis [1,27]. Amino acid metabolites, such as glutamate and aspartate, participate in the regulation of cell death by affecting intracellular metabolic status and signal transduction [28]. Fluctuations of Ca^2+^ during metabolic processes can impact cell survival and death, including necroptosis [29]. Adenosine affects cell signaling pathways, including those related to necroptosis, through its receptors [30]. Metabolic enzymes, such as phosphofructokinase, can influence cellular metabolic status and thus affect the occurrence of necroptosis [31]. Cells undergo metabolic reprogramming to adapt to changes in metabolic products, and the accumulation or deficiency of these products can cause metabolic stress, activating stress response pathways such as adenosine 5′-monophosphate-activated protein kinase (AMPK), thereby affecting cell sensitivity to necroptosis.

## 3. Classic Signaling Pathway of Necroptosis

Although necroptosis could be triggered by various external factors, it is clear that MLKL emerged as an executioner of necroptosis via interacting with the RHIM, which is a conserved protein domain about 18~22 amino acids in length [32,33]. The importance of RHIM interactions for the promotion of necroptosis signaling is underlined by the finding that while the kinase activity of RIPK3 is dispensable for MLKL activation under RIPK3-overexpressed conditions, the RIPK3 RHIM-deficient mutant cannot induce necroptosis even though it still phosphorylates MLKL. The RHIM-dependent RIPK3 oligomer recruits MLKL to the necrosome. In humans, four proteins carrying RHIM domains have been identified: RIPK1, RIPK3, ZBP1, and TRIF (Figure 2) [34].

### 3.1. RIPK1/RIPK3/MLKL-Mediated Necroptosis

RIPK1, a 74 kDa protein, is composed of a N-terminal kinase domain, an intermediate domain (containing the RHIM), and a C-terminal death domain [35]. RIPK1 kinase plays an essential role in mediating deleterious responses downstream of TNFR1, and it is a critical target for the treatment of Parkinson’s disease, Alzheimer’s disease, stroke, lysosomal storage disease, and cancer [36]. In 2000, it was first discovered that the protein kinase RIPK1 plays a crucial role as a kinase-activity-dependent regulator in the induction of necrotic cell death triggered by TNF-α, TRAIL, FasL, tumor necrosis factor-like cytokine 1A (TL1A), or amyloid precursor protein (APP) [9]. Now, RIPK1–RIPK3–MLKL has become the core necroptotic pathway (Figure 3). It is activated by the binding of its substrate, RIPK1, which triggers the phosphorylation of RIPK3 and its subsequent activation of downstream targets. RIPK3 then recruits and phosphorylates MLKL, which is essential for necroptosis to occur.

In the majority of cells, TNFR1 stimulation does not induce cytotoxicity. Instead, it initiates direct pro-inflammatory signaling through the assembly of a membrane-associated protein complex known as complex I (made up of TNF receptor-associated death domain (TRADD), RIPK1, TRAF2, IAP1, IAP2, and linear ubiquitin chain assembly complex (LUBAC)) [37]. Generally, TNFR1 triggers apoptosis via cytosolic complexes Ila (made up of TRADD, FADD, and Casp8) and Ilb (made up of RIPK1, RIPK3, FADD, and Casp8) in sensitized cells. However, necroptosis can be executed through the formation of the complex IIc or the necrosome (made up of RIPK1, RIPK3, and MLKL) under specific conditions or in certain cell types. Signaling toward TNF-induced necroptosis is actively restrained by multiple brakes acting on RIPK1. The presence or absence of RIPK1, along with its post-translational modifications such as ubiquitylation and phosphorylation, plays a critical role in determining the biological outcome. RIPK1 manifests distinct forms within its four different complexes (I, IIa, IIb, and IIc), and their induction occurs dynamically following TNF binding to TNFR1 [38]. Upon the binding of TNF-α to TNFR1, RIPK1 is recruited and undergoes phosphorylation at specific serine residues. The activation of RIPK1, facilitated by its phosphorylation, subsequently triggers the activation of RIPK3 and MLKL kinases, leading to the formation of Complex IIc, eventually culminating in the induction of necroptosis.

Besides TNFR-1, a few RIPK1 inhibitors, such as TNF receptor-associated factor 6 (TRAF6) [39], Necrostain-1 [40], and dimethyl fumarate [41], have also shown an inhibition of the RIPK1/RIPK3/MLKL necroptosis signaling pathway via regulating the abundance of RIPK1. For example, TRAF6 directly interacts with RIPK1 via the polyubiquitination of Lys48-linked RIPK1, thus promoting the proliferation of colorectal cancer cells [39]. These inhibitors may provide a new therapeutic target for specific cancer treatment.

### 3.2. TRIF/RIPK3/MLKL-Mediated Necroptosis

TLRs were the first PRRs to be identified for sensing pathogen-associated peptidoglycan (TLR2), double-stranded (ds)RNA (TLRs), lipopolysaccharide (LPS) (TLR4), flagellin (TLR5), unmethylated CpG DNA motifs (TLR9), and other pathogen-associated molecular patterns [42]. TLRs recruit Toll/IL-1R (TIR) domain-containing adapters to activate gene expression via transcription factors, such as NF-κB and IRF3/IRF7. TLR3 and TLR4 are unique in employing the adapter TRIF to signal. TLR3 and TLR4 drive RIP3 activation directly via the adapter protein TRIF to trigger necroptosis [43]. TRIF is the sole adapter protein for TLR3, but TLR4 signals can be mediated by either TRIF or Myd88 (myeloid differentiation primary response gene). The interaction between TRIF and RIPK3 is associated with the process of necroptosis.

When in the presence of a pancaspase inhibitor, necroptosis would be induced by TLR3 or TLR4 via the TRIF-RIPK3-MLKL axis. Xiaodong Wang and cooperators reported that when TLR3/TLR4 are activated by poly (I:C) and LPS, necrotic death will happen due to the interaction between TRIF and RIP3 through their RHIM domains [15]. TLR3/TLR4-mediated necrosis is independent from both NF-κB activation and IRF3-dependent immune response. Although the kinase activity of RIPK1 is dispensable for TLR3-induced necroptosis in fibroblasts, it is required for TLR3-induced necroptosis in macrophages.

### 3.3. ZBP1/RIPK3/MLKL-Mediated Necroptosis

ZBP1 is an IFN-inducible protein that binds double-stranded Z-form DNA and RNA [44]. ZBP1 is predicted to act as a homeostatic harbor for RHIM-containing RIP kinases where RHIM-dependent functions of RIPK1 restrain the ZBP1-mediated activation of RIPK3. ZBP1 acts as an adapter molecule, connecting RIPK1 to its downstream effectors, such as RIPK3 and MLKL. The interaction of ZBP1 and MLKL is essential for the induction of necroptosis. Studies have shown that ZBP1 can bind to the N-terminal domain of MLKL and activate its kinase activity. Furthermore, ZBP1 is able to recruit MLKL to the cell membrane, where it induces the formation of pores and leads to necroptosis.

Cell death induced by murine cytomegalovirus (MCMV) infection involves a complex formed by ZBP1 and RIPK3, exhibiting RHIM-dependent characteristics while being entirely detached from RIPK1, NF-κB, and interferon signaling. ZBP1 was identified as a cytoplasmic DNA sensor and shown to activate the interferon regulatory factor (IRF) and NF-κB transcription factors, leading to type-I interferon production. Interacting protein RHIM in the DAI protein sequence relays DAI-induced NF-κB signals through the recruitment of the RHIM-containing kinases RIPK1 and RIPK3 [45].

### 3.4. IFNR/MLKL-Mediated Necroptosis

Compared to the promoters (RIPK1, TRIF, and ZBP1), interferon (alpha and beta) receptor 1 (IFNAR1) and interferon-gamma receptor 1 (IFNGR1) can also induce necroptosis in a few cell types [46]. The understanding of the connection between necroptosis and IFNR signaling is still in its infancy. When an IFNR signal was activated via JAK kinases and relevant cognate STAT proteins, a transcriptional program would activate to generate necrosome modulator proteins [47]. Signaling by IFNAR is proposed to contribute to RIPK3 activation in RIPK1-deficient mice because the deletion of *Ifnar1* delays perinatal lethality by several days in mice lacking both RIPK1 and TNFR1. Furthermore, IFN-mediated necroptosis has been reported in cells deficient in RIPK1 or Casp8 [48].

## 4. Necroptosis and Cancer Immunotherapy

### 4.1. Necroptosis and Immune Microenvironment

The necroptosis of tumor cells can regulate and reshape the tumor immune microenvironment (Figure 4), including affecting tumor angiogenesis, the extracellular matrix (ECM), microbial metabolic balance, and immune cell regulation [38,49]. The release of cellular contents due to necroptosis may contain pro-angiogenic factors, such as vascular endothelial growth factor (VEGF) and fibroblast growth factors (FGFs), thereby promoting tumor angiogenesis [50]. Pro-inflammatory cytokines released, such as TNF-α and interleukins (ILs), can also promote angiogenesis by activating endothelial cells [51]. Furthermore, necroptosis may affect the maturity and stability of tumor blood vessels; unstable vessels can lead to hypoxia and uneven nutrition inside the tumor, which in turn affects tumor growth and response to treatment. Enzymes released by necroptosis, such as matrix metalloproteinases, can degrade the ECM, thereby altering the structure and function of the tumor microenvironment, providing conditions for the invasion and metastasis of tumor cells [52]. The microbiome in the tumor microenvironment, including bacteria, can indirectly regulate the survival, metastasis, and drug resistance of tumor cells through their metabolic products. The death of necroptotic cells may change the composition and metabolic activity of the microbiome, thereby affecting the metabolic balance of the tumor microenvironment [53].

In addition, the inflammatory environment induced by necroptosis can significantly affect the infiltration and activation of immune cells [38]. Damage-associated molecular patterns (DAMPs) released by necroptotic cells, serving as signaling molecules, can be recognized by pattern recognition receptors (PRRs) on immune cells, such as TLRs, thereby activating immune cells and promoting their infiltration into the tumor microenvironment [54]. Meanwhile, metabolites released by necroptotic cells, such as ATP, uric acid, and extracellular DNA, can activate immune cells, such as dendritic cells (DCs), promoting antitumor immune responses [55]. The death of necroptotic cells can lead to the production of inflammatory mediators, such as TNF-α, IL-6, IFN-γ, and interleukin-1β (IL-1β), which can further activate and recruit immune cells [56]. In addition, the granulocytes in the TME, especially N1 neutrophils, could produce a large quantity of ROS and reactive nitrogen species (RNS) to induce DNA damage in necroptotic tumor cells [57]. In the tumor microenvironment, necroptosis can affect the polarization state of macrophages. For example, M2 macrophages protect against acute-on-chronic liver failure by inhibiting the necroptosis-S100A9-necroptosis-inflammatory axis, suggesting that M2 macrophages may regulate inflammatory responses and the polarization of immune cells by inhibiting necroptosis [58]. Moreover, mast cells play an anti-inflammatory immunological role during tumor necroptosis by interacting with anti-TFN cytokines or histamine antagonists [59]. Similarly, targeting cell recognition via NK cell receptor ligand interaction and the formation of the immunological synapse could promote NK cell activation. Upon activation, NK cells exert their effector functions, granule exocytosis, or expression of death ligands, inducing tumor cell killing by necroptosis [60].

### 4.2. Necroptosis and Cancer Immunosurveillance

Cancer immunosurveillance is a crucial concept in the field of cancer biology, referring to the body’s natural defense mechanisms, particularly the immune system, in identifying and eliminating cancerous cells [61]. The process involves various components of the immune system, such as DCs, cytotoxic T cells, natural killer (NK) cells, macrophage natural killer T (NKT) cells, and their corresponding cytokines (Figure 5) [62].

For example, RIPK3 regulates the cytokine expression in DCs via a separate, necroptosis-independent pathway because *Ripk3*^−/−^ BMDCs were highly defective in the LPS-induced expression of inflammatory cytokines [63]. In addition, necroptosis plays a regulatory role in the antigen-induced proliferation of T cells. It has been demonstrated that the loss of RIPK3 can rescue the defective T cell proliferation of caspase 8^−/−^ mice [64]. Jiahuai Han and coauthors reported that RIPK signaling and NF-κB expression within the necroptotic cells are critical for the cross-priming of CD8^+^ T cells [65]. Necroptotic sarcoma cells are highly immunogenic, promoting an immune response and reinstating immunosurveillance. Necroptosis may help regulate the intratumoral infiltration of immune-promoting CD8^+^ T cells, activated memory CD4^+^ T cells, and monocytes while reducing the monocytes. Simultaneously, M0 and M2 macrophages, which are favorable prognosis factors, are associated with a sarcoma [49]. The NK cell cytotoxicity is closely related to granzyme-induced necroptosis by ROCK-mediated blebs and/or Rac1-mediated lamellipodia [66]. The inhibition of granzyme B and key necroptosis regulators will attenuate NK cell cytotoxicity. Meanwhile, RIPK3 regulates NKT cell function and promotes the NKT cell-mediated antitumor immune response by activating the mitochondrial phosphatase phosphoglycerate mutase 5 (PGAM5). RIPK3-mediated PGAM5 activation would promote the dephosphorylation of dynamin-related protein 1 and expression of cytokines by facilitating the nuclear translocation of the nuclear factor of activated T cells (NFAT), indicating that RIPK3-PGAM5 signaling would mediate the crosstalk between immune signaling and mitochondrial function [67].

Tumor cells may utilize necroptosis to evade immune surveillance through various mechanisms, including the suppression of immunogenic signal release, the establishment of an immunosuppressive microenvironment, the blocking of antigen presentation, the direct killing of immune cells, metabolic reprogramming in the tumor microenvironment, and the formation of physical barriers [68]. Although necroptosis is generally considered an immunogenic form of cell death, tumor cells may regulate the signaling pathways of necroptosis to reduce the release of immunogenic signals, thereby inhibiting the activation and infiltration of immune cells. Tumor cells may use molecules released during necroptosis, such as transforming growth factor-β (TGF-β), interleukin-10 (IL-10), etc., to establish an immunosuppressive microenvironment, which helps tumor cells escape immune surveillance [51,69]. Tumor cells may downregulate the expression of major histocompatibility complex (MHC) molecules, reducing antigen presentation and making tumor cells unrecognizable to the immune system. Certain molecules (like CCL2) released during necroptosis may have toxic effects on immune cells, leading to the death of immune cells and thus weakening the immune system’s ability to attack tumors [70]. Tumor cells may affect the function of immune cells by changing their metabolic state during necroptosis, for example, by producing immunosuppressive metabolites to inhibit the antitumor activity of immune cells. Furthermore, tumor cells may form a physical barrier around the tumor by secreting molecules such as collagen, preventing immune cells from entering the tumor area [71].

### 4.3. Induced Necroptosis and Immunogenic Cell Death

Immunogenic cell death (ICD) is a distinct form of RCD that is capable of triggering an adaptive immune response targeted toward antigens from dying cells, whether they are endogenous (cellular) or exogenous (viral) [70]. So far, six DAMPs have been mechanistically linked to the perception of RCD as immunogenic: (1) calreticulin (CALR), (2) ATP, (3) high-mobility group box 1 (HMGB1), (4) type I interferon (IFN), (5) cancer cell-derived nucleic acids, and (6) annexin A1 (ANXA1) [72]. In contrast to the passive immune response of cells dying by apoptosis, cells dying by necroptosis actively release DAMPs, cytokines, and chemokines, leading to an inflammatory response. Instead, induced necroptosis is a recently discovered form of programmed cell death triggered by signals from the immune system (Table 1) [38,73]. Injecting cells with activated RIPK3, either in a prophylactic vaccine setting or directly into tumors, promotes CD8^+^ T-cell responses and reduces tumor growth.

Numerous strategies have been developed to regulate necroptosis in cancer cells to facilitate cancer cell death. These strategies include the use of necroptosis inducers such as TNF-α, TRAIL, and FasL, as well as necroptosis inhibitors, such as Necrostatin-1, GSK’872, GSK 872, and necrosulfonamide [83,84,85,86]. In addition, antibodies targeting components of the necroptosis pathway, such as MLKL, have been developed and are currently being evaluated in clinical trials [87,88]. Additionally, chemotherapy agents like mitoxantrone (MTX) and shikonin (SKN) can reinstate anticancer immunosurveillance by inducing the ICD of tumor cells [89]. For example, by the co-delivery of SKN and FePd bimetallic nanozyme via liposomes, ROS-boosting enhanced necroptosis could be obtained to improve therapeutic efficacy via activating immune responses [55]. Furthermore, combined phototherapy based on functional nanomaterials could effectively induce the necroptosis of tumor cells and trigger host immunity by activating antigen-specific T cells [56]. In the clinic, classic necroptosis inducers (TNF + cycloheximide + zVAD, TNF + 5Z-7-oxozeaenol + zVAD, and TNF + IAP inhibitors + zVAD) have been widely used for almost all common cancer types, including head and neck carcinoma, glioblastoma, colorectal cancer, hematopoietic neoplasm, hepatocarcinoma, lung cancer, breast cancer, and leukemia [90].

Induced necroptosis could also be employed to prepare the tumor vaccine. For example, Dmitri V Krysko et al. reported a FADD-dependent RIPK3 induction system to vaccinate cancer cell necroptosis for DAMP release, cytotoxic T cell cross-priming, and IFN-γ production, demonstrating the efficient vaccination potential of immunogenic necroptotic cells [91]. Gabriele Niedermann et al. demonstrated that clinical cisplatin could trigger RIPK3-dependent cell necroptosis and the release of cytosolic mitochondrial DNA, activating the cyclic GMP–AMP synthase (cGAS)–stimulator of interferon genes (STING) pathway to promote T-cell cross-priming by DCs [92]. Jun Chen and coauthors reported a nano-size “artificial necroptotic cancer cell” (αHSP70p-CM-CaP) as a flexible vaccine platform for efficient lymph mode trafficking, multi-epitope-T cell response, IFN-γ-expressing CD8^+^ T cells and natural killer group 2 member D positive (NKG2D+) NK cell expansion [93]. Moreover, by inducing in situ cell necroptosis in the tumor microenvironment via nanomedicine, in situ vaccination could be obtained by triggering DAMP release to promote the maturation of BMDCs, cross-priming of effector T cells, and subsequent cytotoxic effects [94]. In some cases, the activation of necroptosis may have a synergistic effect with immune checkpoint inhibitor therapy, enhancing the ability of immune cells to attack tumors [95].

## 5. Conclusions

As a form of programmed cell death with strong immunogenic characteristics, necroptosis is now established as an essential pathway for cancer treatment. Due to its unique performance to avoid apoptotic resistance and trigger DAMPs to activate the immune response, the current data raise the hope that manipulating necroptosis could provide new and urgently needed therapeutic opportunities in acute and chronic inflammatory conditions. Utilizing diverse drugs, compounds, and agents to target necroptosis, and inducing or manipulating the necroptotic pathway, has emerged as an innovative strategy to overcome apoptosis resistance and bolster antitumor immunity in the realm of cancer therapy.

Nonetheless, many questions remain to be addressed. What are the main molecular targets in the caspase-independent regulatory mechanisms of necroptosis? How does necroptosis contribute to the initiation, amplification, and chronicity of inflammation? What are the main mechanisms to overcome immune drug resistance? Thus, further research in the near future needs to be explored to promote the practical applications of induced necroptosis in cancer immunotherapy.

Insight into the molecular mechanisms of necroptosis: The molecular mechanism is the prerequisite and foundation for cancer treatment. Thus, more efforts should focus on the study of molecular mechanisms, including the activation of key signaling pathways, the role of regulatory factors, and the crosstalk with other forms of cell death.

Novel biomarkers to detect the necroptotic response: As a method for immunogenic cell death, induced necroptosis may effectively synergize with cancer immunotherapy. Therefore, it is of significance to identify and validate biomarkers that can predict the response to necroptosis in order to achieve precision medicine and personalized treatment.

Nanomedicine for precise cell necroptosis induction: Cell necroptosis cannot naturally occur when apoptosis resistance happens. On the contrary, necroptosis can be effectively induced by treatment with nanodrugs or nanoagents. The development of necroptotic drugs will greatly overcome the limitations of current cancer chemotherapy and improve patient outcomes.

Preclinical and clinical studies for practical application: At present, the majority of investigations into therapeutics targeting necroptosis rely on in vitro experiments and/or animal models. Consequently, the clinical feasibility of utilizing specific compounds and anticancer agents requires evaluation through in vivo studies and clinical trials. Furthermore, a comprehensive assessment of the off-target effects associated with necroptosis-targeting therapeutics is essential. Innovative strategies that integrate necroptosis inducers with tumor-targeting agents should be developed to enhance safety and selectivity in clinical applications.

## Figures and Tables

**Figure 1 ijms-25-10760-f001:**
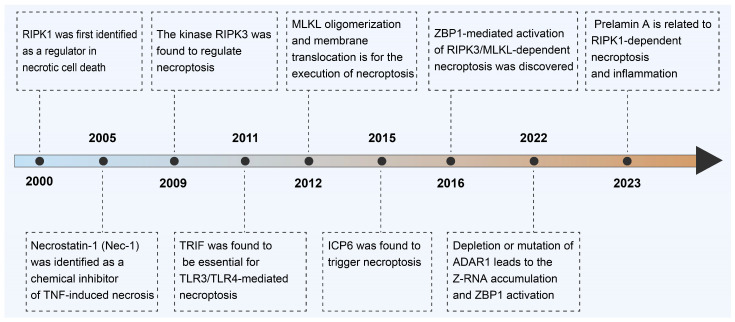
The timeline records the development of necroptosis for cancer treatment. RIPK1, receptor-interacting protein kinase 1; RIPK3, receptor-interacting protein kinase 3; MLKL, mixed-lineage kinase domain-like protein; ZBP1, Z-DNA binding protein 1; TRIF (also called TICAM-1), TIR domain-containing adapter-inducing interferon-β; TLR3, Toll-like receptor 3; TLR4, Toll-like receptor 4; ADAR1, adenosine deaminase acting on RNA enzyme-1.

**Figure 2 ijms-25-10760-f002:**
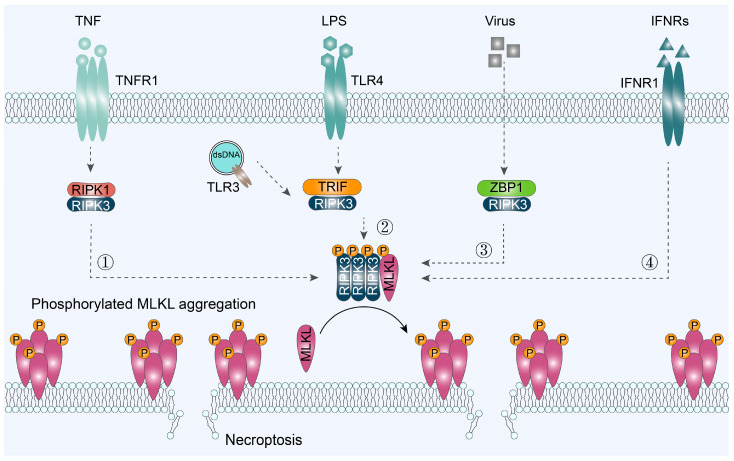
Signaling pathways to trigger cell necroptosis. The induced necroptosis is summarized into four signaling pathways: ① RIPK1/RIPK3/MLKL-, ② TRIF/RIPK3/MLKL-, ③ ZBP1/RPK3/LKKL-, and ④ IFNR/MLKL-mediated necroptosis. TNF, tumor necrosis factor; TNFR1, tumor necrosis factor receptor 1; LPS, lipopolysaccharide; TLR3/4, Toll-like receptors 3 and 4; IFNRs, type I/II interferon receptors; IFNR1, interferon alpha and beta receptor subunit 1.

**Figure 3 ijms-25-10760-f003:**
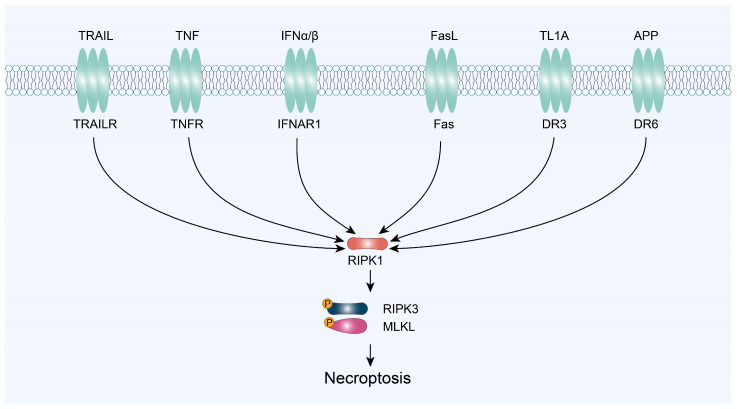
Signaling pathways to trigger cell necroptosis via RIPK1/RIPK3/MLKL axis. TRAIL, TNF-related apoptosis-inducing ligand; TNF, tumor necrosis factor; IFNα/β, interferon α/β; FasL, Fas ligand; TL1A, tumor necrosis factor-like cytokine 1A; APP, amyloid precursor protein.

**Figure 4 ijms-25-10760-f004:**
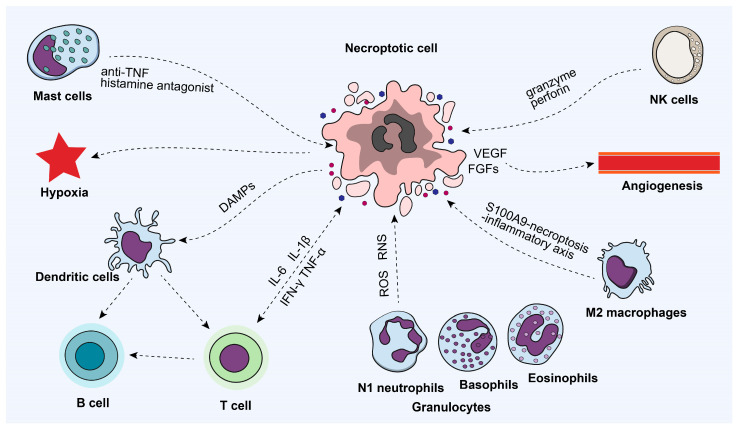
The immune microenvironment associated with tumor necroptosis. VEGF, vascular endothelial growth factor; FGFs, fibroblast growth factors; DAMPs, damage-associated molecular patterns.

**Figure 5 ijms-25-10760-f005:**
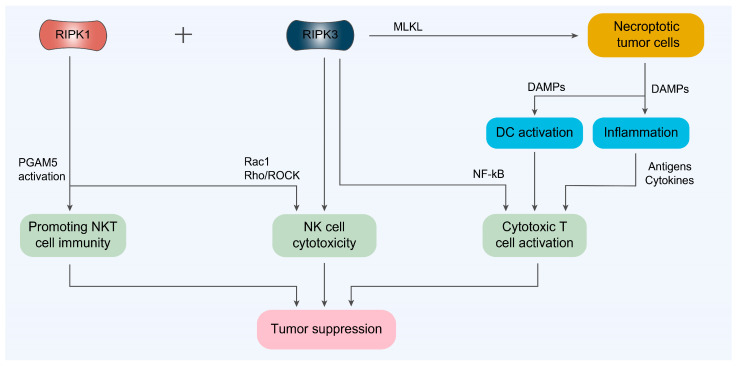
The relationship between necroptosis and cancer immunosurveillance. NF-κB, nuclear factor kappa-B; NKT cell, natural killer T cell.

**Table 1 ijms-25-10760-t001:** Induced tumor necroptosis by various agents for enhanced immunotherapy.

Agents	Cancer Type	Pathway	Ref.
Lytic dead cell	B16F10 melanoma	BATF3+ cDC1- and CD8^+^ leukocyte-dependent	[74]
5-FU/IDN-7314	hHT-29 Colon cancer	NF-κB- and RIP1-mediated necroptosis	[75]
TSZ/mitoxantrone	mTC-1 Lung carcinoma	RIPK3/MLKL	[76]
Poly(l:C)/zVAD-fmk	mCT-26 Colon carcinoma	Modulating the tumoricidal microenvironment and dendritic cell-inducing antitumor immune system	[77]
Polymeric Nanobubbles	CT26 Colon carcinoma	Sonoimmunotherapy-mediated maturation of dendritic cells and activation of CD8^+^ cytotoxic T cells	[78]
MLKL- and tBid-mRNA	B16F10 melanoma	Type I interferon signaling and Batf3-dependent dendritic cells	[79]
BP-bPEI-PEG/CpG	4T1 Breast cancer	Trigger the release of damage-associated molecular patterns to potentiate the immune response	[80]
FePd/SKN@Lip	4T1 Breast cancer	Trigger the release of damage-associated molecular patterns to potentiate the immune response	[55]
2-methoxy-6-acetyl-7-methyjuglone	HCT116, HT29, A549 cells and cisplatin-resistant A549 cells	Lysosomal membrane permeabilization, mitochondrial dysfunction, ROS production	[81]
Oncolytic virus/mitoxantrone	U2OS cells and TUBO cells osteosarcoma	Revealing pro-inflammatory cytokine production and myeloid cells and cytotoxic T cell influx in local and distant tumors	[82]

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
