# Peer review of "Induced Necroptosis and Its Role in Cancer Immunotherapy"

_ijms, 2024, doi:10.3390/ijms251910760_

Round 1

Reviewer 1 Report

Comments and Suggestions for Authors

The perspective of the review writing on “Induced Necroptosis and Its Role in Cancer Immunotherapy” is notable and has insightful knowledge. This review is well-structured having an interesting context about four signaling pathways of necroptosis; RIPK1-RIPK3-mediated necroptosis, TRIF-RIPK3-mediated necroptosis, ZBP1-RIPK3-mediated necroptosis, and IFNRs-mediated necroptosis and their immunotherapeutic role in cancer.

I have a few minor suggestions for the author, please clarify and corrected these errors. 

1.     Author needs to correct and rewrite the sentence line 294 to 299. 

2.     The conclusion contains repetitive ideas, and the third paragraph is unclear. Please rewrite the third paragraph to avoid redundancy and ensure clarity. Also, the conclusion is too general. The author needs to revise it to make it more specific and contextual to the topic.

3.     Minor typological errors are noticed in the manuscripts please correct them before final submission.

4.     Author efforts are observed, to input a lot of information presented in an essence summarized manner. This review has very impressive writing skills and is well-linked. Overall, the manuscript is informative, nice, and ready to publish after minor revision.  

Author Response

Reviewer #1

The perspective of the review writing on “Induced Necroptosis and Its Role in Cancer Immunotherapy” is notable and has insightful knowledge. This review is well-structured having an interesting context about four signaling pathways of necroptosis; RIPK1-RIPK3-mediated necroptosis, TRIF-RIPK3-mediated necroptosis, ZBP1-RIPK3-mediated necroptosis, and IFNRs-mediated necroptosis and their immunotherapeutic role in cancer.

I have a few minor suggestions for the author, please clarify and corrected these errors. 

  1. Author needs to correct and rewrite the sentence line 294 to 299. 

Reply: Many appreciations for kind suggestion. The sentences have been revised as follows: “Necroptotic sarcoma cells are highly immunogenic, promoting an immune response and reinstating immuno-surveillance. necroptosis might may help regulate intratumoral infiltration of immune-promoting CD8+ T cells, activated memory CD4+ T cells, and monocytes while reducing the monocytes, while simultaneously M0 and M2 macrophages, which are favorable prognosis factors are associated with a sarcoma.[47] The NK cell cytotoxicity is closely related to granzyme-induced necroptosis by ROCK-mediated blebs and/or Rac1-mediated lamellipodia.[61]

  1. The conclusion contains repetitive ideas, and the third paragraph is unclear. Please rewrite the third paragraph to avoid redundancy and ensure clarity. Also, the conclusion is too general. The author needs to revise it to make it more specific and contextual to the topic.

Reply: Thank you very much for valuable advices. We carefully revised the conclusion section (divided into key opinions) to be more specific and contextual. “Insight into the molecular mechanisms of necroptosis”, “Novel biomarkers to detect the necroptotic response”, “Nanomedicine for precise cell necroptosis inducing”, and “Preclinical and clinical studies for practical application”

  1. Minor typological errors are noticed in the manuscripts please correct them before final submission.

Reply: Thanks a lot for nice comment. We carefully checked the typological errors and revised them.

  1. Author efforts are observed, to input a lot of information presented in an essence summarized manner. This review has very impressive writing skills and is well-linked. Overall, the manuscript is informative, nice, and ready to publish after minor revision.

Reply: Many thanks for your kind efforts. We are trying our best to revise the manuscript and offer guidelines for readers.

Reviewer 2 Report

Comments and Suggestions for Authors

Section 2 is more like history of necroptosis discovery than the hallmarks of necroptosis. Authors could look into that

Page 3, paragraph 2: statements in line118-120 and 125-127 are missing references

Page 4, paragraph 1: from the narrative presented it is actually not clear that MLKL is the executioner of the necroptosis. Authors may consider revising appropriately

Figure 2 has several unnamed cartoons, and it is not clear which on is MLKL

Page 4, line 163: no need to repeat abbreviations after defining them at the time of introduction

Figure 3 is not representative of the text discussed. Also it is sort of contradicting figure 2. Finally, from the figure it is not clear if FADD, RPK1 and Caspase-8 are bound to each other as a complex or acting independently

Authors should use consistent terminology throughout the manuscript. For example either TRIF should be used or TICAM-1. Using both can confuse the reader

From the article the checkpoints that control necroptosis and cell death are not clear. For example, TLR and IFNR pathways are well known regulators of immune cell activity and if they can lead to necroptosis, then reader would be interested in understanding what is preventing cell death upon activation of these pathways

Page 7: Figure 4 is not correlating with the text

Page 9, line 325: Authors could give examples for “certain molecules”

Author Response

Reviewer #2

  1. Section 2 is more like history of necroptosis discovery than the hallmarks of necroptosis. Authors could look into that.

Reply: Many thanks for kind recommendation. We changed the subtitle into “Milestone on Discovery of Necroptosis”.

  1. Page 3, paragraph 2: statements in line118-120 and 125-127 are missing references.

Reply: Thanks a lot for nice advices. The references have been added. “Br J Cancer 2023, 129, 1707–1716” and “Apoptosis 2023, 28, 1304–1314”

  1. Page 4, paragraph 1: from the narrative presented it is actually not clear that MLKL is the executioner of the necroptosis. Authors may consider revising appropriately.

Reply: Many appreciations for valuable suggestion. We revised the sentence to avoid the misunderstand. Although necroptosis could be triggered by various external factors, it is clear that MLKL emerged as an executioner of necroptosis via interacting with the RHIM……

  1. Figure 2 has several unnamed cartoons, and it is not clear which on is MLKL.

Reply: Thank you very much for kind reminding. We added the name on the top of cartoons to increase the readability.

  1. Page 4, line 163: no need to repeat abbreviations after defining them at the time of introduction.

Reply: Many thanks for kind comment. We deleted the repeated abbreviations.

  1. Figure 3 is not representative of the text discussed. Also it is sort of contradicting figure 2. Finally, from the figure it is not clear if FADD, RPK1 and Caspase-8 are bound to each other as a complex or acting independently.

Reply: Many appreciations for nice advices. For the four signaling pathways of necroptosis, RIPK1/RIPK3/MLKL-mediated one is the most classic and important. Therefore, Figure 3 is presented to summarize all possible pathways to trigger RIPK1. The Figure has been simplified to increase the readability.

  1. Authors should use consistent terminology throughout the manuscript. For example either TRIF should be used or TICAM-1. Using both can confuse the reader.

Reply: Thanks a lot for nice reminding. We have unified them using TRIF.

  1. From the article the checkpoints that control necroptosis and cell death are not clear. For example, TLR and IFNR pathways are well known regulators of immune cell activity and if they can lead to necroptosis, then reader would be interested in understanding what is preventing cell death upon activation of these pathways.

Reply: Many appreciations for your valuable comments. TLRs have a crucial role in the detection of microbial infection in mammals and insects. And the specific function depends on the target, Toll-interacting proteins or adaptors. Recognition of microbial products by Toll-like receptors expressed on dendritic cells triggers functional maturation of dendritic cells and leads to initiation of antigen-specific adaptive immune responses (left images, Ref: Nat Rev Immunol 2001, 1, 135–145). However, TLR signaling activates RIPK3 and induces necroptosis, leading to the death of infected cells to protect the host during viral or microbial infection (right images, Ref: Pediatr Res 2022, 91, 73–82).

  1. Page 7: Figure 4 is not correlating with the text.

Reply: Thanks a lot for kind reminding. The contents of Figure 4 have been revised and the relationship between necroptosis and immune microenvironment has been systematically discussed according to Figure 4.

  1. Page 9, line 325: Authors could give examples for “certain molecules”.

Reply: Many thanks for kind advice. We have revised the sentence to make it more clear. “Certain molecules (like CCL2) released during necroptosis may have toxic effects on immune cells, leading to the death of immune cells and thus weakening the immune system's ability to attack tumors.[69]

Reviewer 3 Report

Comments and Suggestions for Authors

General: This manuscript attempts to describe different mechanisms of necroptosis, and the effect that necroptosis-inducers have on the success of immunotherapy. However, the description of the different necroptotic pathways is confusing and should be better depict in a scheme/figure (please see comments 4-6). The effects on the activation of the immune system and on immunotherapy are not well described as described in the comments below.  

Major comments:

1.       The abbreviation should be defined when first mentioned, and not later.  There are numerous places, including the abstract where this happens.  Sometime only the abbreviation appears and nothing is defined later on (for example ERE, line 96)

2.       Complex IIc is not defined and appears only once (line 96). 

3.       Some proteins are mentioned but their functions and relations to necroptosis are not clear. For example prelamin A, ADAR1.

4.       Figure 2:  the different complexes that are formed in the TRIF and ZBP1 are not clear. Is the necrosome only RIPK1/RIPK3/MLKL is presented. However, is RIPK1 also included in the complex when the TRIF pathway or the ZBP1 pathway are activated? If not - three complexes should appear in the figure. Importantly, the different regulators of the process, mention in lines 110-134, should be incorporated into the figure. 

5.       Figure 3 and in the text – Fas and CD95 are different names for the same protein. Please do not mention them as separate entities. 

6.       Lines 169-184 and figure 3: the decision-making leading to necroptosis is described, but in a confusing manner. The ubiquitination and phosphorylation of RIPK1 are major factors in this process and should be described better. What is the difference between complex IIa and IIb? What is complex IIc (line 177)? The conditions leading to necroptosis require that caspase 8 is inhibited, but this is hardly mentioned, only in passing, and should be greatly elaborated: what inhibits caspase-8 in the different RIPK1/RIPK3/MLKL pathway, and is it similarly inhibited in the TRIF or ZBP1 pathways?

7.       Lines 266-279:  the order that events take place is not well structured. First cell die by apoptosis, and release DAMPs. Nearby macrophages take up the DAPMs that change their polarization, and then, as a result, they secrete pro-inflammatory cytokines. The way it is written now does not emphasize the correct order. Moreover, pro-inflammatory cytokines are not limited to IL-1β or TNFα, and anti-tumoral cytokines, such as IFNβ are more important (please see PMID 33205451). 

8.       Figure 4: The figure should include how necroptotic cells affect the (polarization) of immune cells, and only then how the immune microenvironment affect tumor cells (and not necessarily necroptotic tumor cells, as these are already dead). How does RIPK3 activate NF-κB? How does RIPK1 affect Rho GTPase? Do neutrophils release N2O or NO2?

9.       Lines 313-332: how do these general mechanisms that allow tumors to evade the immune system related specifically to necroptosis? 

10.   Lines 341-342 - An important comment that needs to be elaborated: why is it that spontaneous necroptosis is not sufficient to trigger and anti-tumoral response? Why do we need to intervene in order to see such a positive effect?   

11.   Line 354: Why or how do Necrostatin-1, GSK'872, GSK 872, and necrosulfonamide that inhibit necroptosis used to induce necroptosis? This is very confusing.

12.   Lines 363-366: if the necroptosis inducers are used in the clinics, what are their objective rates of success (Overall survival, progression-free survival)?

 Minor Comments: 

1. Line 149 – the complex ZBP1/RPK3/LKL should be ZBP1/RPK3/MLKL. 

2.  Line 300 – how does granzyme trigger necroptosis?   

Comments on the Quality of English Language

English is fine

Author Response

Reviewer #3

General: This manuscript attempts to describe different mechanisms of necroptosis, and the effect that necroptosis-inducers have on the success of immunotherapy. However, the description of the different necroptotic pathways is confusing and should be better depict in a scheme/figure (please see comments 4-6). The effects on the activation of the immune system and on immunotherapy are not well described as described in the comments below.  

Major comments:

  1. The abbreviation should be defined when first mentioned, and not later.  There are numerous places, including the abstract where this happens.  Sometime only the abbreviation appears and nothing is defined later on (for example ERE, line 96)

Reply: Many appreciations for nice comment. We carefully checked the manuscript and unified the abbreviations when they are fist appear.

  1. Complex IIc is not defined and appears only once (line 96). 

Reply: Thanks a lot for kind reminding. The puzzle has been clarified.

  1. Some proteins are mentioned but their functions and relations to necroptosis are not clear. For example prelamin A, ADAR1.

Reply: Many appreciations for valuable suggestion. The function of protein ADAR1 has been discussed in line 95-98 when it first appears. ADAR1, a major repressor of immune responses activated by endogenous retroviral elements , would lead to the Z-RNA accumulation and ZBP1 activation, which culminated in RIPK3-mediated necroptosis.[19] Text for prelamin A is at line 100-101. “…..defective process of prelamin A will trigger nuclear RIPK1-dependent necroptosis and inflammation.

  1. Figure 2:  the different complexes that are formed in the TRIF and ZBP1 are not clear. Is the necrosome only RIPK1/RIPK3/MLKL is presented. However, is RIPK1 also included in the complex when the TRIF pathway or the ZBP1 pathway are activated? If not - three complexes should appear in the figure. Importantly, the different regulators of the process, mention in lines 110-134, should be incorporated into the figure. 

Reply: Sorry for the puzzle by the error in Figure 2. The phosphorylated RIPK3/MLKL complex has been revised. Besides, the discussion in section 2 (lines 110-134) mainly focus on the intrinsic regulation process of tumor necroptosis. However, Figure 2 aims to distinguish the four typical necroptotic pathways.

  1. Figure 3 and in the text – Fas and CD95 are different names for the same protein. Please do not mention them as separate entities. 

Reply: Thanks a lot for kind suggestion. The Fas and CD95 have been unified throughout the manuscript.

  1. Lines 169-184 and figure 3: the decision-making leading to necroptosis is described, but in a confusing manner. The ubiquitination and phosphorylation of RIPK1 are major factors in this process and should be described better. What is the difference between complex IIa and IIb? What is complex IIc (line 177)? The conditions leading to necroptosis require that caspase 8 is inhibited, but this is hardly mentioned, only in passing, and should be greatly elaborated: what inhibits caspase-8 in the different RIPK1/RIPK3/MLKL pathway, and is it similarly inhibited in the TRIF or ZBP1 pathways?

Reply: Many appreciations for your valuable advice. TNFR1-induced necroptosis will lead to the formation of four receptor-bound complex (I, IIa, IIb, and IIc/necrosome) (Ref: Nature 2015, 517, 311–320). Among of them, only complex IIc or the necrosome (made up of RIPK1, RIPK3 and MLKL) formation will induce RIPK1 kinase activity and RIPK3-kinase-activity-dependent necroptosis. We have revised the discussion to distinguish the function of four complex. Meanwhile, the contents of Figure 3 have been revised to improve the readability. “……Generally, TNFR1 triggers apoptosis via cytosolic complexes Ila (made up of TRADD, FADD and Casp8) and Ilb (made up of RIPK1, RIPK3, FADD and Casp8) in sensitized cells. However, necroptosis can be executed through the formation of the complex IIc or the necrosome (made up of RIPK1, RIPK3 and MLKL) under specific conditions or in certain cell types…….

  1. Lines 266-279:  the order that events take place is not well structured. First cell die by apoptosis, and release DAMPs. Nearby macrophages take up the DAPMs that change their polarization, and then, as a result, they secrete pro-inflammatory cytokines. The way it is written now does not emphasize the correct order. Moreover, pro-inflammatory cytokines are not limited to IL-1β or TNFα, and anti-tumoral cytokines, such as IFNβ are more important (please see PMID 33205451). 

Reply: Thank you very much for kind suggestions. The discussion about Figure 4 has been revised to march the contents in figure, and the relationship between DAMPs and macrophage polarization has also been clarified.The release of DAMPs by necroptotic cells will serve as signaling molecules to activate immune cells and promote their infiltration into the tumor microenvironment (line 265-268). M2 macrophages protect against acute-on-chronic liver failure by inhibiting the necroptosis-S100A9-necroptosis-inflammatory axis, suggesting that M2 macrophages may regulate inflammatory responses and the polarization of immune cells by inhibiting necroptosis.”

  1. Figure 4: The figure should include how necroptotic cells affect the (polarization) of immune cells, and only then how the immune microenvironment affects tumor cells (and not necessarily necroptotic tumor cells, as these are already dead). How does RIPK3 activate NF-κB? How does RIPK1 affect Rho GTPase? Do neutrophils release N2O or NO2?

Reply: Thanks a lot for kind reminding. The contents of Figure 4 have been revised and the relationship between necroptosis and immune microenvironment has been systematically discussed point-by-point according to Figure 4.

  1. Lines 313-332: how do these general mechanisms that allow tumors to evade the immune system related specifically to necroptosis? 

Reply: Many thanks for your advice. In revised manuscript, we discussed the relevant mechanism about how tumor evade immune system via necroptosis. “Tumor cells may use molecules released during necroptosis, such as transforming growth factor-β (TGF-β), interleukin-10 (IL-10), etc., to establish an immunosuppressive microenvironment, which helps tumor cells escape immune surveillance.[51,69] Tumor cells may downregulate the expression of major histocompatibility complex (MHC) molecules, reducing antigen presentation and making tumor cells unrecognizable to the immune system. Certain molecules (like CCL2) released during necroptosis may have toxic effects on immune cells, leading to the death of immune cells and thus weakening the immune system's ability to attack tumors.[70]

  1. Lines 341-342 - An important comment that needs to be elaborated: why is it that spontaneous necroptosis is not sufficient to trigger and anti-tumoral response? Why do we need to intervene in order to see such a positive effect?

Reply: Many appreciations for your valuable comment. We deleted the comment in order to make mistake.

  1. Line 354: Why or how do Necrostatin-1, GSK'872, GSK 872, and necrosulfonamide that inhibit necroptosis used to induce necroptosis? This is very confusing.

Reply: Thank you very much for kind advice. The discussion about necroptosis inducer and inhibitor have been revised. Necrostatin-1, GSK'872, GSK 872, and necrosulfonamide are necroptosis inhibitors, not inducer.

  1. Lines 363-366: if the necroptosis inducers are used in the clinics, what are their objective rates of success (Overall survival, progression-free survival)?

Reply: Many thanks for kind reminding. As a researcher, it is hard to predict the objective rate of overall survival for necroptotic treatment. Take an example, Tsukasa Seya reported that a combined regimen of polyI:C and zVAD induced approximately 50% CT26 necroptosis in vitro without secondary effects of TNFα or type I IFNs (Ref: Cancer Immunol Res (2015) 3 (8): 902–914). Meanwhile, injection of 100 μg polyI:C resulted in >80% regression by day 20 of CT26-implanted tumors compared with injection of a PBS control, and tumoricidal action of 1 mg zVAD only was negligible

 Minor Comments: 

  1. Line 149 – the complex ZBP1/RPK3/LKL should be ZBP1/RPK3/MLKL. 

Reply: Thanks a lot for reminding. We have revised the typo.

  1. Line 300 – how does granzyme trigger necroptosis?

Reply: Sorry for puzzle. We have revised the text to explain the mechanism. “Upon activation, NK cells exert their effector functions, granule exocytosis, or expression of death ligands, inducing tumor cell killing by necroptosis.

Round 2

Reviewer 3 Report

Comments and Suggestions for Authors

The authors have sufficiently revised the manuscript. I have no additional comments.  

Comments on the Quality of English Language

 Minor English editing is needed.